# Effect of an External Magnetic Field on the Hydrogen Reduction of Magnetite Nanoparticles in a Polymer Matrix

**Petr Chernavskii** [1,2,3], **Sveta Ozkan** [2], **Galina Karpacheva** [2], **Galina Pankina** [1,2,3] and **Nikolai Perov** [4,*]

1 Department of Chemistry, Lomonosov Moscow State University, 119991 Moscow, Russia; chern5@inbox.ru (P.C.); pankina5151@inbox.ru (G.P.)
2 A.V. Topchiev Institute of Petrochemical Synthesis, Russian Academy of Sciences, Leninsky Prospect, 29, 119991 Moscow, Russia; ozkan@ips.ac.ru (S.O.); gpk@ips.ac.ru (G.K.)
3 Institute of Organic Chemistry N.D. Zelinsky, RAS, 119991 Moscow, Russia
4 Faculty of Physics, Lomonosov Moscow State University, 119991 Moscow, Russia
* Correspondence: perov@magn.ru; Tel.: +7-495-939-1847

**Abstract:** A hybrid electromagnetic nanomaterial, which is a matrix based on a conjugated polymer of poly-3-amine-7-methylamine-2-methylphenazine with dispersed magnetite nanoparticles immobilized on multi-walled carbon nanotubes, has been synthesized. In situ magnetometry was used to study the kinetics of the hydrogen reduction of $Fe_3O_4$ immobilized in the structure of a ternary nanocomposite in magnetic fields of different intensities. An increase in the magnetite reduction reaction rate with the formation of metallic iron nanoparticles at $T = 420\,°C$ and at a magnetic field strength in the range of 60–3000 Oe was observed. The dependence of the degree of conversion of $Fe_3O_4$ on the magnetic field strength was established.

**Keywords:** magnetite nanoparticles; hydrogen reduction; conjugated polymer matrix; multi-walled carbon nanotubes; ternary nanocomposite

## 1. Introduction

Nanotechnology and its products are playing an increasingly important role in the new century. The prospect of large quantities of nanoparticles in a wide variety of materials is opening the way to new types of industrial production based on them. Therefore, nanoparticles play an essential role not only in nature and modern science, but also in advanced technologies. The increased interest of researchers in nanoparticles is due to their unique size-dependent chemical and physical properties and, consequently, the high potential for their practical use. Today, the unique physical properties of nanoparticles are being studied intensively, with particular attention being paid to magnetic nanoparticles. The magnetic properties of such nanoparticles most clearly demonstrate the differences between bulk and nanomaterials. Magnetic nanoparticles are widespread in nature, and they can be found in many biological objects. They are used in medicine, in information recording and storage systems, magnetic cooling systems, as magnetic sensors, etc. [1–5]. Intensive research is carried out both in the field of synthesis of magnetic nanoparticles and in the discovery of possibilities of their practical application [6,7]. The study of chemical reaction patterns during the synthesis of such particles, the dependence of these patterns on external conditions, and the mechanisms of chemical transformations are objects of concentrated attention of scientists all over the world. Many methods have been developed for such studies [8–10]; however, most are based on a comparison of the analysis results of the initial and the final reaction products [11,12]. There was no direct control of the magnetization of the reaction medium during the reaction. At the same time, some reactions involving reagents with magnetic order and a significant magnetic moment allow to follow the course of the reaction, registering changes in the magnetic properties of the corresponding reagents in real time [13]. The additional data obtained in this way opens

up new possibilities for explaining the mechanisms of magnetic field influence on chemical reactions, which have been widely studied for a long time [14–16]. Moreover, this method makes it possible to study the influence of a magnetic field on the kinetics of reactions with magnetic particles [17,18].

The rate of topochemical reactions, among other things, depends significantly on both the particle size and shape, as well as the defectiveness of the particle surface. The latter determines the number of potential nucleation centers of a new phase. Both the particle size and its defectiveness are determined by the way of nanoparticle synthesis, their environment, and may vary over a wide range.

Previously, the influence of an external magnetic field on the kinetics of hydrogen reduction of metal oxides was studied. The first study of the reaction kinetics of hydrogen reduction of magnetite in magnetic fields of different intensities showed that the reaction mechanism strongly depends on the external magnetic field strength [17]. In [18], the effect of a magnetic field on the kinetic parameters of hydrogen reduction of cobalt oxide $Co_3O_4$ under isothermal conditions was studied. When a bulk sample of $Co_3O_4$ was studied, the effect of the magnetic field on the activation energy of the reduction reaction was found, whereas in the case of $Co_3O_4$ particles immobilized on silica, this effect was not observed. The difference in the reduction kinetics of cobalt oxide is related to the effect of the magnetic field on the structure of defects in a solid sample, whose concentration decreases as the size of the $Co_3O_4$ particles decrease.

As magnetic nanoparticles become nano-sized, their surface energy rises sharply, resulting in a strong tendency to aggregate. One of the most common ways to prevent the aggregation of magnetic nanoparticles is to immobilize them on the surface of a carbon material or to stabilize them in a polymer matrix. In recent years, particular interest has been shown in ternary hybrid multifunctional nanocomposites, which include conjugated polymers, magnetite nanoparticles, and carbon nanomaterials [19–23]. Of the carbon nanomaterials of interest are carbon nanotubes, which are hollow cylindrical structures with diameters ranging from a tenth of a nanometer to several tens of nanometers. They are characterized by high mechanical strength and flexibility at low density, and high thermal and electrical conductivity. The potential of a composite material based on nanocomponents with magnetic and electrical properties is of great interest. Such nanocomposite materials, as new generation materials, are characterized by improved functional properties. In the present work, the kinetics of hydrogen reduction of magnetite nanoparticles immobilized on multi-walled carbon nanotubes (MWCNT) dispersed in a matrix of poly-3-amine-7-methylamine-2-methylphenazine (PAMMP) is studied for the first time using in situ magnetometry. The effect of the magnetic field on the reduction rate even in weak fields and the dependence of the $Fe_3O_4$ conversion degree on the strength of an external magnetic field have been shown.

## 2. Experimental

### 2.1. Materials

Standard reagents were used in the synthesis. 3-Amine-7-dimethylamine-2-methyl-phenazine hydrochloride (ADMPC), $FeSO_4 * 7H_2O$ and $FeCl_3 * 6H_2O$ (all high-purity grade), aqueous ammonia, acetonitrile, and DMF (all reagent grade from Acros Organics, Geel, Belgium) were used without further purification. Ammonium persulfate $((NH_4)_2S_2O_8)$ (analytical grade) was purified by recrystallization from distilled water. Aqueous solutions of the reagents were prepared using distilled water.

### 2.2. Synthesis of $Fe_3O_4$/MWCNT/PAMMP

To prepare the $Fe_3O_4$/MWCNT/PAMMP nanocomposite [19], the $Fe_3O_4$/MWCNT nanomaterial synthesis was first performed by hydrolysis of $FeSO_4 * 7H_2O$ (0.86 g) and $FeCl_3 * 6H_2O$ (2.35 g) at a 1:2 ratio in a solution of ammonium hydroxide (3.21 g) [24] in the presence of MWCNT (MER, United States) at 60 °C. The mass of carbon nanotubes was $C_{MWCNT}$ = 3 wt.% of the weight of ADMPC (0.0114 g) which was equal to 0.000342 g.

Then, to synthesize the $Fe_3O_4$/MWCNT/PAMMP nanocomposite, the freshly prepared $Fe_3O_4$/MWCNT nanocomposite was added directly into the ADMPC solution in acetonitrile (0.02 mol/L, 0.38 g). Then, an aqueous solution (30 mL) of ammonium persulphate (0.04 mol/L, 0.548 g) was added dropwise to the $Fe_3O_4$/MWCNT/ADMPC suspension in acetonitrile. The volume ratio of organic to aqueous phase was 1:1 ($V_{total}$ = 60 mL). The synthesis was continued for 4 h with intensive stirring at 15 °C [19]. At the end of the synthesis, the mixture was precipitated in a five-fold excess of distilled water. The resulting product was filtered off, washed repeatedly with distilled water to remove residual amounts of reagent, and dried over KOH under vacuum to constant weight. The yield of $Fe_3O_4$/MWCNT/PAMMP was 1.16 g at $C_{Fe}$ = 45.9% (according to ICP-AES data).

*2.3. Magnetic Measurements*

The test sample of 10 mg mass was placed in a microreactor which simultaneously served as the measuring cell of a vibrating sample magnetometer [13]. To avoid the nanoparticle movement the magnetite powder was fixed between two gas-permeable porous quartz membranes. Each test sample was heated to 400 °C in an Ar flow of 30 mL/min and held at this temperature for 30 min to remove the adsorbed water. It was then cooled to a predetermined temperature and the Ar flow was replaced by a flow of $H_2$ at a rate of 30 mL/min. The volumetric flow rate was $6 \times 10^4\,h^{-1}$, which allowed us to exclude the effect of external diffusion inhibition. To control the amount of magnetic component, a small magnetic field, on the order of 60 Oe, is switched on and the induced magnetic moment is measured. The magnitude of the signal from the sample is proportional to the mass of the magnetic component and the field strength. If the amount of magnetic component changes during the reaction, the magnetometer's signal value will also change proportionally. In this way, it is possible to monitor the amount of reacted substance in real time. The time dependence of the magnetization was measured at a frequency of 1 Hz at fixed values of the magnetic field strength. The kinetic dependence of the magnetization on time was, thus, obtained at a given temperature. The amount of the magnetic phase was obtained from the dependence of the magnetization on the magnitude of the magnetic field strength by extrapolation to zero field. By making measurements at different magnetic field strengths its effect on the parameters of the chemical reaction can be assessed.

*2.4. Material Characterization*

The metal content in the nanocomposite was measured quantitatively by an inductively coupled plasma atomic emission spectroscopy method (ICP-AES) using a Shimadzu ICP emission spectrometer (ICPE-9000) (Kyoto, Japan).

The XRD study was performed in ambient atmosphere using a Difray-401 X-ray diffractometer (Scientific Instruments Joint Stock Company, Saint-Petersburg, Russia) with Bragg–Brentano on $CrK_\alpha$ radiation, $\lambda$ = 0.229 nm focusing geometry with a scan step 0.02 degrees, and scan rate 0.01 s/step in the angular range $2\theta$ = 20–120°. The results of the XRD analysis were used to calculate the size distribution of the coherent scattering regions of the crystallites in the magnetic nanoparticles.

An electron microscopic study was carried out using a LEO912 AB OMEGA transmission electron microscope (Bioz Inc., Los Altos, CA, USA) and a Hitachi TM 3030 scanning electron microscope (Hitachi High-Technologies Corporation, Fukuoka, Japan) with up to 30,000 magnification and 30 nm resolution. Prior to the analysis, the sample was thoroughly ground in an agate mortar in ethanol and left for a few hours. A drop of the supernatant was then applied to a polymer film placed on a copper grid. The particle size distribution was obtained by visual analysis of the micrographs using "Image-Pro Plus 6.0" software. A histogram was constructed considering 600–1000 particles.

FTIR spectra were measured in air on a Bruker IFS 66v FTIR spectrometer (Karlsruhe, Germany) in the range of 400–4000 $cm^{-1}$ and analyzed using the Soft-Spectra software. The samples were prepared as KBr pressed pellets.

## 3. Results

### 3.1. Synthesis and Characterization of the Fe₃O₄/MWCNT/PAMMP Nanocomposite

The $Fe_3O_4$/MWCNT/PAMMP hybrid nanomaterial was obtained under the conditions of in situ oxidative polymerization of ADMPC in an aqueous solution of acetonitrile in the presence of $Fe_3O_4$/MWCNT [19]. The formation of the $Fe_3O_4$/MWCNT/PAMMP hybrid nanocomposite material includes: synthesis of $Fe_3O_4$/MWCNT by hydrolysis of a mixture of ferrous and ferric salts at a molar ratio of 1:2 in an ammonium hydroxide solution in the presence of MWCNT; fixation of the monomer on the surface of the previously prepared $Fe_3O_4$/MWCNT nanocomposite introduced into the nanomaterial synthesis reaction medium, followed by in situ polymerization of ADMPC in the presence of an aqueous solution of an oxidizing agent.

The formation of the $Fe_3O_4$/MWCNT/PAMMP nanocomposite was confirmed by FTIR spectroscopy, XRD, transmission (TEM) and scanning (SEM) electron microscopy, and inductively coupled plasma atomic emission spectroscopy (ICP-AES).

In the FTIR spectrum of the $Fe_3O_4$/MWCNT/PAMMP nanocomposite, all the main bands characterizing the chemical structure of PAMMP are retained [19]. The splitting and the shift of bands at 1609 and 1500 $cm^{-1}$ (Figure 1), corresponding to stretching vibrations of the $\nu$ $c$–$c$ bonds in aromatic rings, indicate the interaction of planar phenazine units of PAMMP with aromatic structures of MWCNT. In the FTIR spectrum of the $Fe_3O_4$/MWCNT/PAMMP nanomaterial, an intense absorption band appears at 572 $cm^{-1}$ corresponding to stretching vibrations of the Fe–O bond, confirming the formation of $Fe_3O_4$.

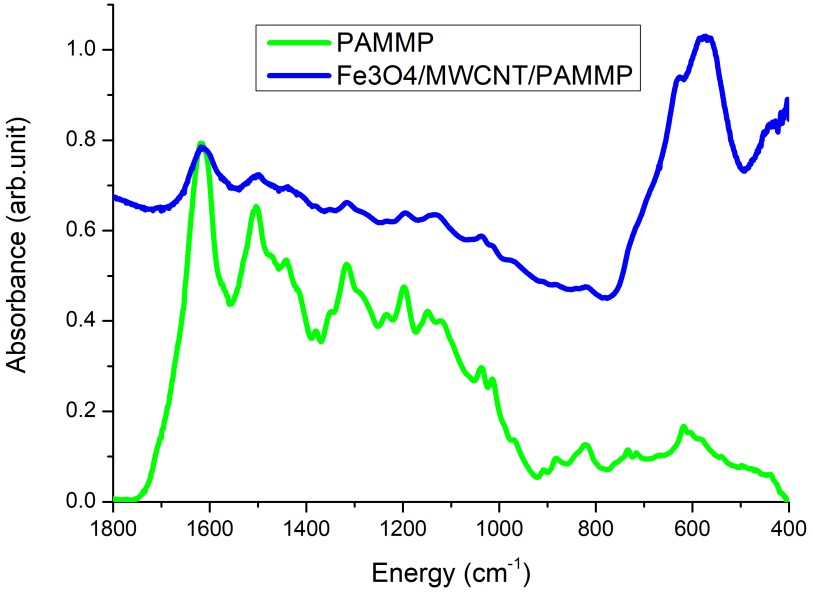

**Figure 1.** FTIR spectra of PAMMP and $Fe_3O_4$/MWCNT/PAMMP.

### 3.2. X-ray Diffraction

The formation of a $Fe_3O_4$-based nanocomposite is confirmed by XRD data. Figure 2 shows the diffraction pattern of the $Fe_3O_4$/MWCNT/PAMMP nanocomposite. The diffraction peaks at scattering angles $2\theta$ = 46.1°, 54.2°, 66.9°, 84.8°, 91.2°, and 102.2° ($CrK_\alpha$-radiation) correspond to $Fe_3O_4$.

These diffraction peaks relate to the Miller indices (220), (311), (400), (422), (511), and (440), and correspond to the cubic structure of $Fe_3O_4$ (JCPDS 19-0629) [25]. The diffraction peak at $2\theta = 39.9°$ corresponds to the carbon phase of MWCNT. Mathematical processing of the diffraction patterns, phase analysis of the samples, and particle size estimation by the Scherer method were performed using the High Score Plus program and the JSCD file cabinet. The average particle size of magnetite was found to be approximately 7.4 nm. The

polymer on the surface of $Fe_3O_4$/MWCNT reduces the aggregation of nanoparticles during the synthesis of $Fe_3O_4$/MWCNT/PAMMP.

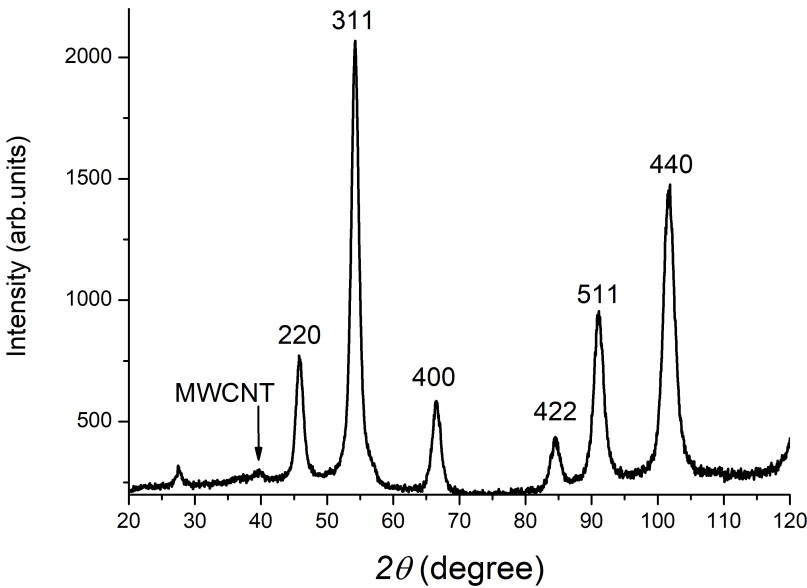

**Figure 2.** XRD of $Fe_3O_4$/MWCNT/PAMMP.

*3.3. Magnetic Characterization*

Figure 3 shows the magnetization of the original sample as a function of the magnetic field at room temperature. Measurements were taken from the maximum winding current (corresponding to approximately 10 kOe) to zero current in the electromagnet. At zero current, the solenoid windings were switched. After switching the contacts with increasing current, first the residual induction of the solenoid core was compensated, then a magnetic field of opposite direction was generated and gradually increased to its maximum value. It should be noted that the change in the sign of the signal, corresponding to the remagnetization of the sample, took place in a small field of the order of 10–20 Oe. Therefore, the coercive force and the residual magnetization are close to zero, indicating the presence of superparamagnetic nanoparticles. The measurement results give a sample saturation magnetization value of 46 em/g. This value was determined by linear interpolation of the linear portion of the measurement curve to zero magnetic field value. It should be noted that the magnetic properties of particles can change considerably when their shape is changed, since the shape anisotropy energy is comparable in magnitude to the magnetocrystalline anisotropy. However, this question requires a separate consideration and we hope to show its role in our further investigations.

The behavior of superparamagnetic non-interacting particles can be described by the Langevin function $M(H, T) = M_S L[mH/k_B T]$, where $M(H, T)$ is the magnetization at temperature $T$ and field $H$, $M_S$ is the saturation magnetization, $L$ is the Langevin function $L(x) = \coth x - 1/x$, and $m = M_S(\pi d^3/8)$ is the magnetic moment of a sphere.

The dependence of the relative magnetization $J/J_S$ of the nanocomposite (where $J$ is the value of the magnetization at a field strength H, and $J_S$ is saturation magnetization) on the ratio of the magnetic field strength H to the temperature T was studied for two temperatures ($T = 289$ K and 390 K). The experimental curves obtained are shown in Figure 4. The uniformity of the curves obtained at different temperatures confirms the presence of superparamagnetic particles. The average superparamagnetic particle diameter can be obtained by fitting the slope of the magnetization near zero field regions $(dM/dH)_{H=0}$ at 300 °K. The following relationship was used for calculation $d = (18k_B T (dM/dH)_{H=0}/\pi \varrho M_S^2)^{1/3}$, where $\varrho$ is the density [26]. The slope, $(dM/dH)_{H=0}$ at 300 °K, of each sample was obtained by separating the contribution from the superparamagnetic phase. The size of the

superparamagnetic phase particles was estimated to be around 4–8 nm. The saturation magnetization $M_S$ of $Fe_3O_4$ was assumed to be 90 emu/g at ambient conditions [26].

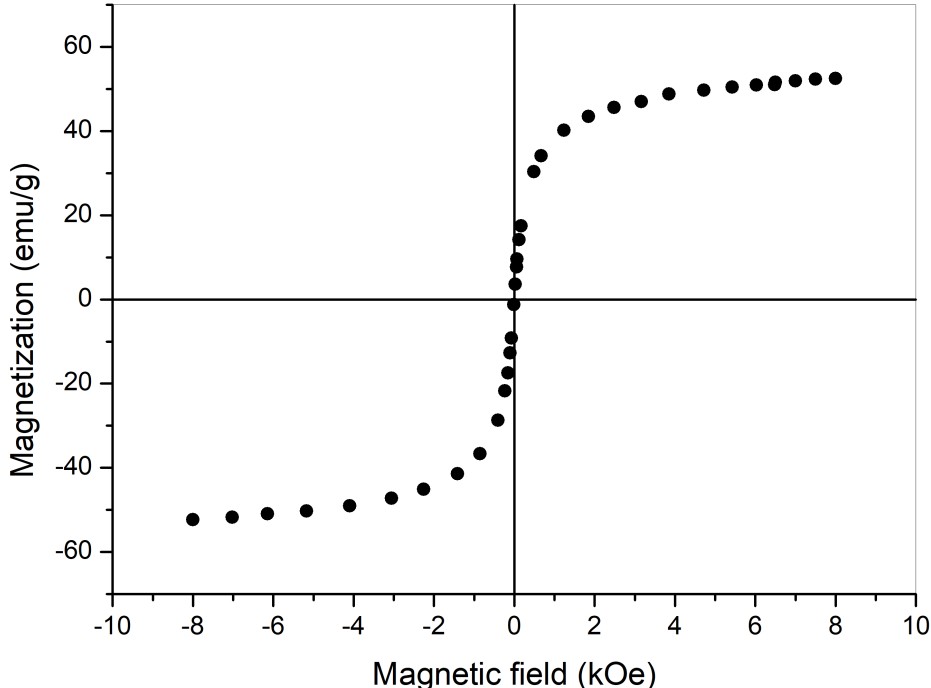

**Figure 3.** Magnetization of $Fe_3O_4$/MWCNT/PAMMP as a function of applied magnetic field at room temperature.

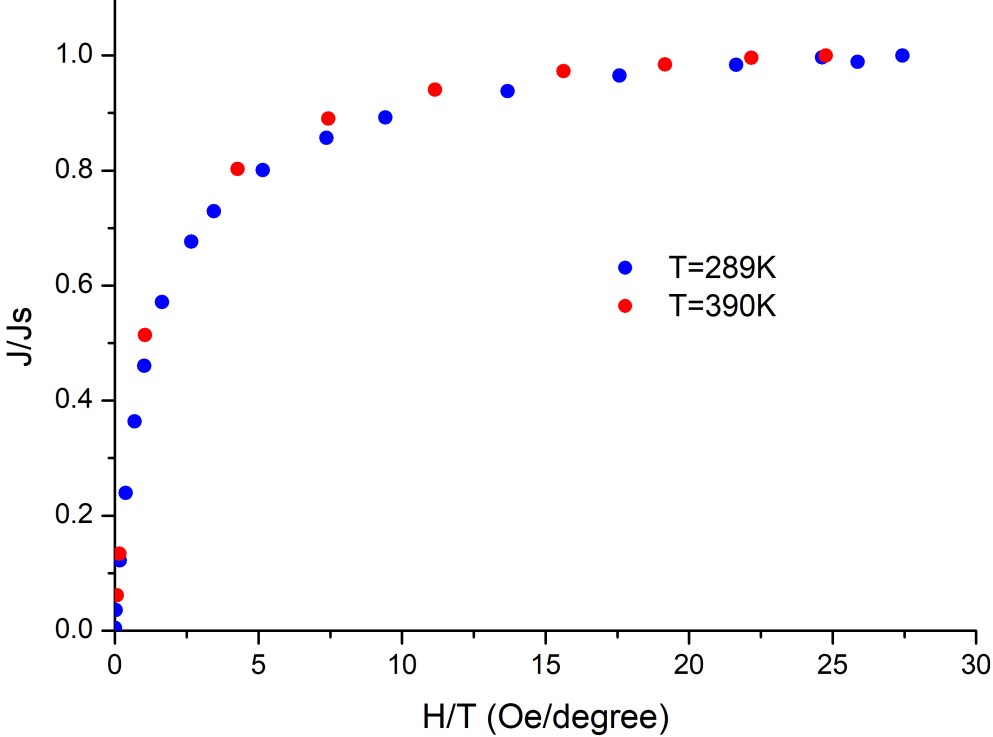

**Figure 4.** Dependence of relative magnetization on the ratio of field strength—temperature. Blue dots—289 °K, red dots—390 °K.

### 3.4. Morphology of Nanocomposites

The course of chemical reactions is greatly influenced by the surface condition of the reagents. Therefore, data on the morphology of the samples are of great interest. As mentioned above, a transmission electron microscope and a scanning electron microscope were used to obtain this information. Figures 5 and 6 show SEM and TEM images of the $Fe_3O_4$/MWCNT/PAMMP nanocomposite. As can be seen, the $Fe_3O_4$ nanoparticles immobilized on the MWCNT surface are distributed in the polymer matrix.

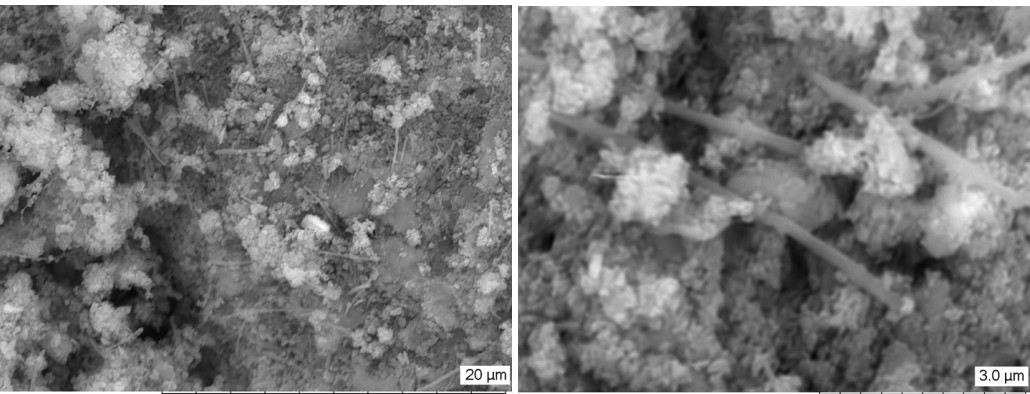

**Figure 5.** SEM image of the typical nanocomposite sample $Fe_3O_4$/MWCNT/PAMMP.

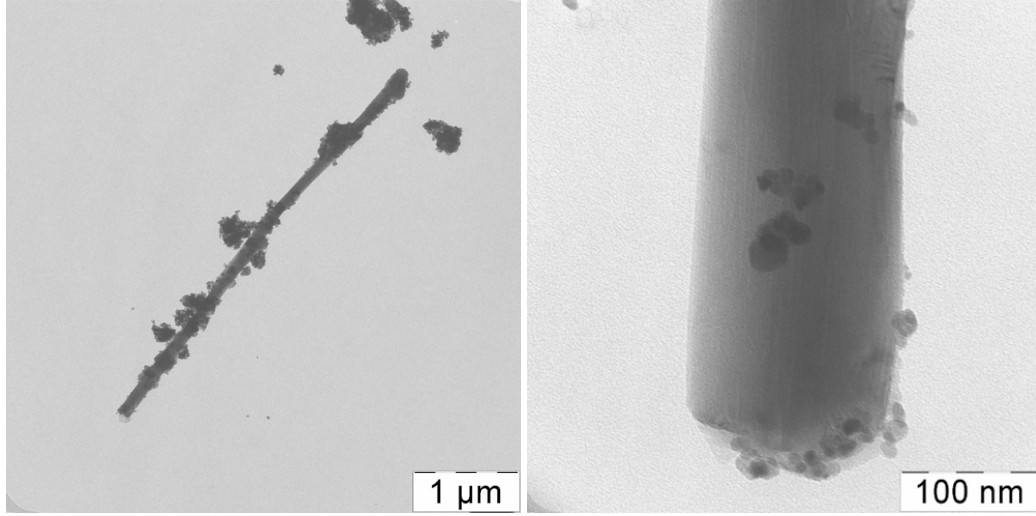

**Figure 6.** TEM image of the typical nanocomposite sample $Fe_3O_4$/MWCNT/PAMMP.

According to the TEM data (Figure 6), the spherical $Fe_3O_4$ nanoparticles are 4–11 nm in size. The size of the nanoparticles is determined using the EsiVision software (eVision Software, The Hague, The Netherlands). The sizes of the magnetite nanoparticles correspond to the XRD data. According to ICP-AES, the content of Fe = 45.9%.

To assess the particle sizes and their changes during the recovery process, photographs of TEMS were taken for which particles of various sizes were counted using the software described above. Typical photos are shown in Figure 7a–c.

Next to each type of particle is a corresponding histogram of the dimensional distribution Figure 7d–f. As can be seen from the above figures, the average particle size of $Fe_3O_4$ in the initial composite is 6.8 ± 0.1 nm, after recovery in the field 60 Oe, the average particle size of Fe is 6.3 ± 0.1 nm, after recovery in the field 1 kOe—4.2 ± 0.1 nm. After recovery in the 1 kOe field, a significant decrease in the average size of the Fe particles is observed. The bright halo around the darker iron particles is caused by the oxidation process of Fe nanoparticles at room temperature. Voids are present at the interface between the oxide shell and the Fe core [27]. It is easy to see that the sizes of the magnetite particles

(dark areas) correspond to the XRD data. Furthermore, the sizes of the particles lead to the superparamagnetic behavior described above.

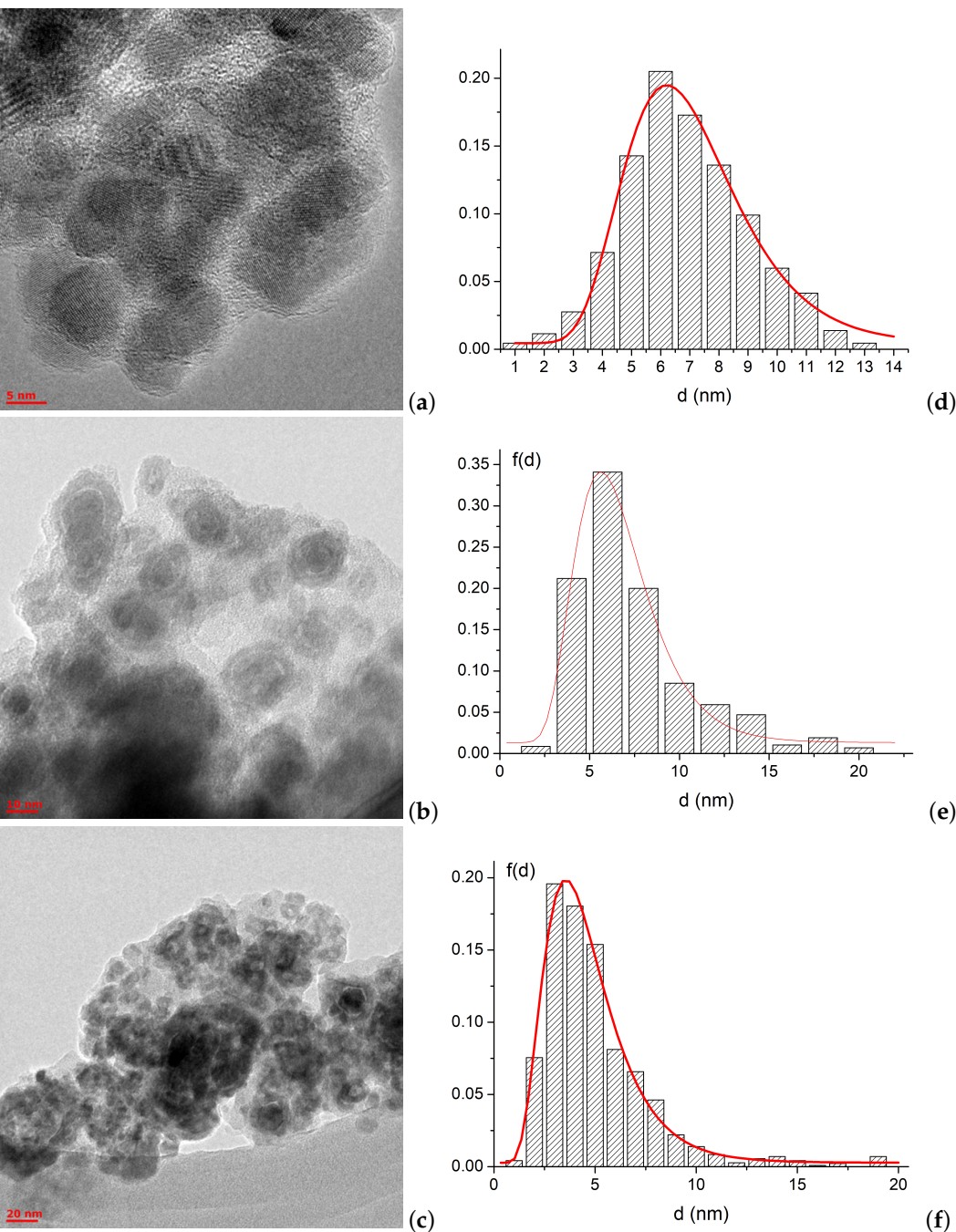

**Figure 7.** TEM image of the nanocomposite samples before and after reduction (left) and the corresponding magnetite particle size distributions (right), (**a**,**d**)—initial; (**b**,**e**)—after recovery in the field 60 Oe; and (**c**,**f**)—after recovery in the field 1 kOe.

### 3.5. Reduction Kinetics of Fe₃O₄/MWCNT/PAMMP with Hydrogen

The reduction of solid oxide phases refers to topochemical reactions occurring at the interface between the reduced oxide and its product. The rate of chemical interactions, as noted above, depends on the state and size of the interface and on the type of reducing agent. According to the classical theory, the reduction mechanism includes two stages (after adsorption of the reducing agent on the reaction surface):

(1) The crystallochemical act involving the transition of oxygen of the oxide lattice to the adsorbed reducing agent molecules with a simultaneous rearrangement of the initial oxide lattice to that of the reduction product;

(2) Desorption of the gaseous reduction product.

A magnetic field can have an effect on each of the above stages. In order to study the mechanisms of magnetic field action the time dependence of magnetic component amount change during reduction reaction was investigated. Figure 8 shows the dependencies of magnetization on time when $Fe_3O_4$/MWCNT/PAMMP was exposed to a current of $H_2$ at T = 420 °C in magnetic fields from 60 Oe to 3 kOe. Here, $J(t)$—magnetization at time $t$ and $J_{t=0}$—initial magnetization. The reduction in magnetization at $t \approx 40$ s is due to the intermediate formation of wustite FeO, which is paramagnetic at the recovery temperature and does not create magnetic moment. In a special series of experiments, the reduction process was stopped when the minimum point on the curve in Figure 8 was reached. When measured, the saturation magnetization at the minimum point on the curves demonstrated a constant value. This indicates the invariability of the amount of magnetite at the time corresponding to the minimum point on the curves. In other words, the amount of magnetite transformed into wustite does not depend on the strength of the external magnetic field. The subsequent increase in magnetization is associated with the formation of metallic iron particles as a result of FeO reduction. At $t > 500$ s, the magnetization remains constant for all values of the external field strength. This indicates the completion of the reduction process and the complete transformation of magnetite into iron.

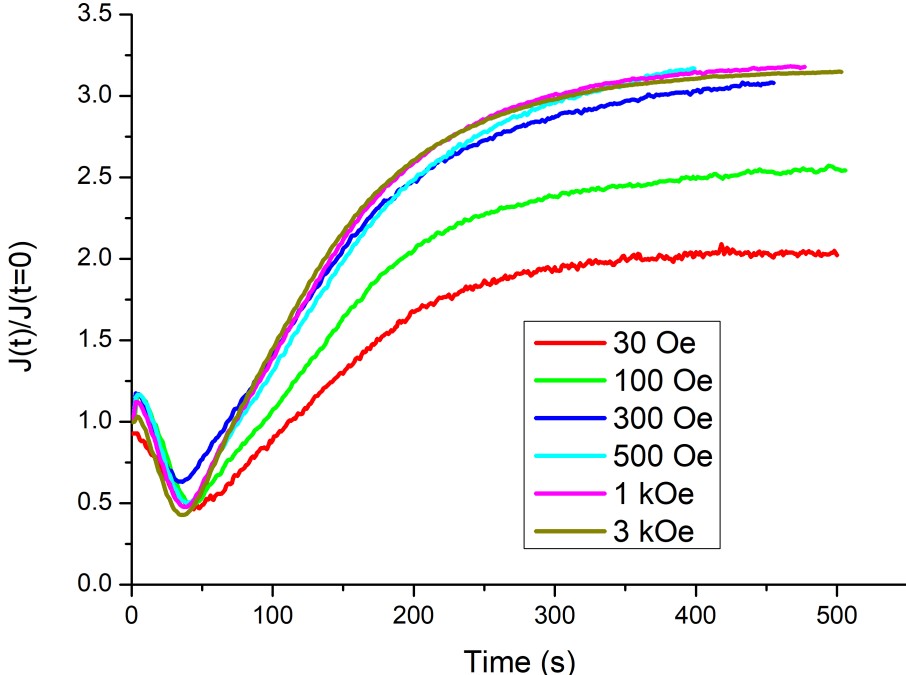

**Figure 8.** Time dependence of relative magnetization $J(t)/J_0$ of samples for different magnetic field values during recovery in hydrogen.

Part of the kinetic curve from the minimum of magnetization to a constant value can be interpreted as the dependence of the degree of transformation on time, where the minimum corresponds to the zero degree of transformation, and the end of the process corresponds to complete conversion to iron. Figure 9 shows the dependence of the degree of transformation on time for various intensities of the external magnetic field.

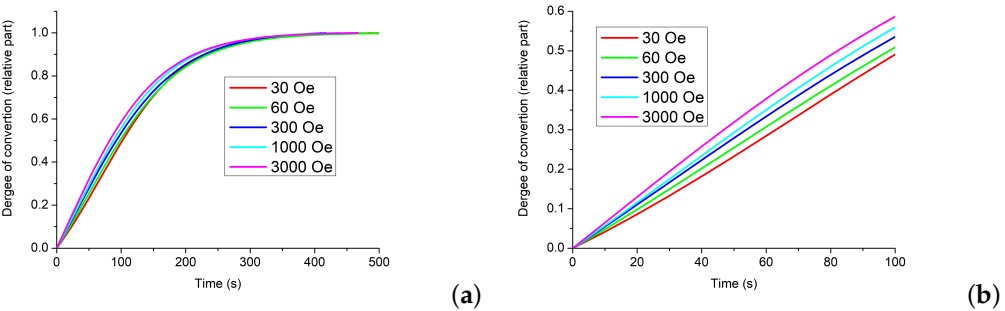

**Figure 9.** Time dependence of relative magnetization $J(t)/J_0$ for various magnetic field value (**a**); initial part $0 \leq t \leq 100$ s (**b**).

The results of kinetic experiments were processed within the framework of the classical theory of topochemical processes [28]. As it was noted above for the vast majority of gas–solid topochemical reactions, the process involves the simultaneous occurrence of two processes, the nucleation of a new phase and the growth of embryos. It is assumed that the rate of nucleation obeys the exponential law

$$\frac{dN(\tau)}{d\tau} = S * z_0 * k_1 * \exp\left(-k_1 * \tau\right) \tag{1}$$

where $z_0$ is the number of potential nucleation centers, $S$ is the specific surface area $k_1$ is the rate constant of nucleation, $N$ is the number of embryos. The embryo growth rate is

$$W(t, \tau) = \frac{4\pi}{3} * k^3 * (t - \tau)^3 \tag{2}$$

where $k$ is the embryo growth rate constant. The reaction rate is

$$r(t) = \int_0^t W(t, \tau) * \frac{dN(\tau)}{d\tau} \, d\tau \tag{3}$$

After the integration and a number of subsequent calculations according to the model mentioned above [28], we obtain the equation for the degree of transformation

$$\alpha(t) = 1 - \exp(-k * (t - \frac{1}{k_1}[1 - \exp(-k_1 * t)])) \tag{4}$$

where $\alpha$—degree of transformation and $t$—time. In Figure 9, the kinetic curves are shown with a high degree of accuracy, described by Equation (4). It was found that the growth rate constant of the embryos did not depend on the magnitude of the applied magnetic field. Figure 10 shows the dependence of the reduction rate (constant $k1$) in relative units on the magnetic field value. It follows that the magnetic field affects the nucleation process and does not affect the growth rate of the embryos. The appearance of the embryos of a new phase is preceded by the dissociative adsorption of hydrogen. We believe that the influence of the magnetic field is associated with this stage of the recovery process.

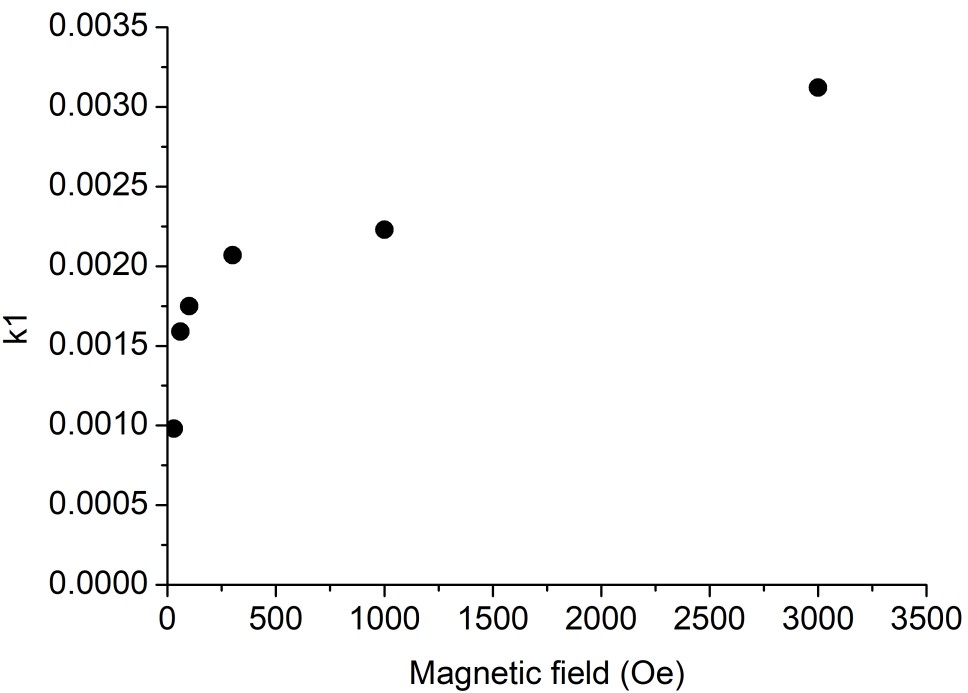

**Figure 10.** Field dependence of the reduction rate.

### 4. Discussion

Previously, we investigated the effect of the magnetic field on the kinetics of the reduction of magnetite nanoparticles in hydrogen [17]. It was shown that the reaction rate changes significantly in a magnetic field above 1 kOe. The observed effect was explained by a change in the adsorption of hydrogen with a change in the spin structure of the magnetic particles. A similar effect was observed in the reduction of cobalt ferrite particles [29]. The results of both works correspond to the theoretical model of Melander et al. [30], in which it is shown that the change in the electronic structure of nanoparticles significantly affects their interaction with hydrogen. Considering the above, this paper has attempted to modify the electronic structure of magnetite nanoparticles through their interaction with carbon nanotubes in the composite. The aim of this work was to study the effect of an external magnetic field on the kinetics of hydrogen reduction of magnetite in a nanocomposite comprising a polymeric matrix in which magnetite nanoparticles immobilized on carbon nanotubes are dispersed. The results obtained confirm the expediency of the chosen approach. Thus, an external magnetic field leads to an increase in the rate of the magnetite reduction in composite with carbon nanotubes. During the reaction, two phases coexist in the system, the magnetite phase and the iron phase so that the field-induced phase transformation by $\Delta G_M$ is a possible mechanism. $\Delta G_M$ of the Gibbs free energy tends to be more stable of a ferromagnetic state with large $m$ and promotes the transition from a ferromagnet with a smaller magnetic moment to a larger one. In our case, this is a transition from a ferrimagnetic to a ferromagnetic with a large magnetic moment. Assuming a simple model with only the Zeeman effect, $\Delta G_M$ under H can be written as

$$\Delta G_M = \mu_0 H \Delta m_S \tag{5}$$

where $\mu_0$ is the permeability of a vacuum and $\Delta m_S$ is the change of the saturation magnetic moment [18]. In present case, $\Delta m_S$ is expressed by

$$\Delta m_S = m(Fe_3O_4) - m(Fe) \tag{6}$$

where $m(Fe_3O_4)$ and $m(Fe)$ are the magnetic moments for the magnetite and iron phas-es, respectively. The magnetic field effect sign is found by the sign of $\Delta m_S$. Because the

$m(Fe_3O_4) < m(Fe)$ so that the sign of $\Delta m_S$ and $\Delta G_M$ is negative. This indicates that the magnetic field promotes the phase transformation from the $Fe_3O_4$ to Fe phase [31].

## 5. Conclusions

For the first time, we investigated the effect of an external magnetic field on the reduction kinetics, not of free magnetite but of magnetite nanoparticles, which are part of the nanocomposite material. A hybrid nanomaterial based on PAMMP, MWCNT, and magnetite nanoparticles was synthesized by in situ chemical oxidative polymerization of ADMPC in the presence of $Fe_3O_4$/MWCNT nanocomposite. The method of in situ magnetometry was used to study the reduction kinetics of magnetite nanoparticles immobilized on multi-walled carbon nanotubes with hydrogen in different magnetic fields. The rate of $Fe_3O_4$ reduction reaction with hydrogen increases under the influence of an external magnetic field. The dependence of the degree of magnetite conversion on the external magnetic field strength was confirmed. The activation energy for the growth of new phase nuclei decreases with increasing magnetic field strength. Unfortunately, the question of the effect of nanocomposite morphology, which is undoubtedly important, has been left out of consideration; however, it requires a separate, serious study and we hope to show this in our future research.

**Author Contributions:** Conceptualization, P.C., G.K. and N.P.; methodology, P.C. and G.K.; software, P.C.; validation, P.C., G.K. and G.P.; formal analysis, P.C. and N.P.; investigation, P.C., G.P. and S.O.; resources, P.C. and G.K.; data curation, G.P.; writing—original draft preparation, P.C.; writing—review and editing, N.P.; visualization, P.C. and S.O.; supervision, N.P.; project administration, P.C. All authors have read and agreed to the published version of the manuscript.

**Funding:** This research received no external funding.

**Institutional Review Board Statement:** Not applicable.

**Informed Consent Statement:** Not applicable.

**Data Availability Statement:** Not applicable.

**Acknowledgments:** The work was supported in part by M.V. Lomonosov Moscow State University Program of Development, and this work was carried out in part within the State Program of TIPS RAS. N.P. acknowledges support from Russian Ministry of Science and Education, grant No. 075-15-2021-1353.

**Conflicts of Interest:** The authors declare no conflict of interest.

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
