# Peer review of "Effect of an External Magnetic Field on the Hydrogen Reduction of Magnetite Nanoparticles in a Polymer Matrix"

_magnetochemistry, doi:10.3390/magnetochemistry9050123_

Round 1
Reviewer 1 Report (New Reviewer)
The manuscript by Nikolai Perov et al. reported the effect of magnetic on the reduction of Fe3O4 using hydrogen. Overall, the work is very interesting. However, some questions need to be addressed before the acceptance of this work in Magnetochemistry.
1. Figure 5 only has two figures, however, the caption showed TEM images of the nanocomposite samples before and after reduction, (a,d) - initial; (b,e) - after recovery in the field 60 Oe; (c,d) - after recovery in the field 1 kOe.
2. XRD of Fe3O4 after reduction should be provided.
3. The caption of Figure 7 should be detailed for a, b, c, d, e, and f.
4. What are the conditions for different lines in Figure 8.
Author Response
The authors are grateful to the reviewer for reading the article and for the comments made. The present version of the article has been substantially revised and slightly expanded. All typos and errors identified by the reviewers have been corrected during the final reading. The English language was checked by a professional translator.
The following changes to the text were made according to the individual comments, which are marked with colour highlighting.
The introduction has been enlarged and expanded.
The authors did not increase the number of references, but added explanations directly in the text of the paper.
The description of experimental methods has been expanded.
Substantial changes have been made in the presentation of experimental results - some figures have been replaced, comments necessary in the text have been added.
The conclusions are revised slightly.
Reviewer #1.
Remark: The manuscript by Nikolai Perov et al. reported the effect of magnetic on the reduction of Fe3O4 using hydrogen. Overall, the work is very interesting. However, some questions need to be addressed before the acceptance of this work in Magnetochemistry.
Answer: We are grateful for the positive evaluation of the work and have tried to take into account all comments made.
1. Remark: Figure 5 only has two figures, however, the caption showed TEM images of the nanocomposite samples before and after reduction, (a,d) - initial; (b,e) - after recovery in the field 60 Oe; (c,d) - after recovery in the field 1 kOe.
Answer: The caption to figure 5 (figure 6 in the present version) has been corrected.
2. Remark: XRD of Fe3O4 after reduction should be provided.
Answer: We agree with the reviewer's suggestion about the desirability of presenting XRD samples after reduction (they should not contain magnetite, only iron), but we have not been able to do this on the samples we have.
3. Remark: The caption of Figure 7 should be detailed for a, b, c, d, e, and f.
Answer: The caption for figure 7 has been corrected.
4. Remark: What are the conditions for different lines in Figure 8.
Answer: In Figure 8 the values of the magnetic field strength at which the corresponding dependencies were obtained have been added.
Reviewer 2 Report (New Reviewer)
There are terrible problems with the writing and layout of the whole article, and the article is hardly innovative, since Fe3O4/MWCNT complex catalyst has been reported too much. Moreover, there are many incorrect description or overstatement of results. and lack of balance in the introduction and discussion.In addition, the poor quality of figures makes the manuscript unreadable, specifically as follows:1. In Fig.1,The XRD data of the pure sample such as Fe3O4 and MWCNT is missing and the standard cards need to be supplemented. The resolution is another problem.
2. Fig.5 HRTEM of Fe3O4 and MWCNT should be given. The surface elemental states of the sample need to be provided, this can be made through XPS survey spectra and high-resolution of C, Fe, O elements.
3. As mentioned above, the resolution and layout of Figure 7 should be changed and enhanced. The legends are too general and should be more specific, the diagram sequence is not appropriate.In addition, I wonder how the particle size distribution is measured? It appears to be inconsistent with TEM images?
4. The result discussion section is not specific and too rough.
Therefore, given the current quality of the manuscript, I am sorry for rejecting it!
There are serious problems in the writing of the article, and the innovation is insufficient. The resolution of the picture is low, and the authenticity of the data needs to be investigated.
Author Response
The authors are grateful to the reviewer for reading the article and for the comments made. The present version of the article has been substantially revised and slightly expanded. All typos and errors identified by the reviewers have been corrected during the final reading. The English language was checked by a professional translator.
The following changes to the text were made according to the individual comments, which are marked with colour highlighting.
The introduction has been enlarged and expanded.
The authors did not increase the number of references, but added explanations directly in the text of the paper.
The description of experimental methods has been expanded.
Substantial changes have been made in the presentation of experimental results - some figures have been replaced, comments necessary in the text have been added.
The conclusions are revised slightly. We hope that the improved and corrected version of the article will receive a more positive reviewer's evaluation.
Reviewer #2.
Remark: There are terrible problems with the writing and layout of the whole article, and the article is hardly innovative, since Fe3O4/MWCNT complex catalyst has been reported too much. Moreover, there are many incorrect description or overstatement of results. and lack of balance in the introduction and discussion. In addition, the poor quality of figures makes the manuscript unreadable.
Answer: We are grateful to the reviewer for his frank opinion, but want to draw his attention that the main purpose of the work is not just to obtain Fe3O4/MWCNT complex catalyst, but first of all to test the effect of the magnetic field on the recovery rate of magnetite in the composition of the composite. And as it seems to us, our goal was achieved: a significant influence of magnetic field on reduction rate was found, which is confirmed by the data given in the article.
1. Remark: In Fig.1,The XRD data of the pure sample such as Fe3O4 and MWCNT is missing and the standard cards need to be supplemented. The resolution is another problem.
Answer: The main purpose of the diffractograms shown (Figure 2) is to confirm the presence of magnetite without additional iron oxides. The second aim is to estimate the size of the magnetite particles. In our opinion, the data presented in the article provide an answer to both questions. Incidentally, in the text of the article we note the presence of a low amplitude peak corresponding to carbon nanotubes.
2. Remark: Fig.5 HRTEM of Fe3O4 and MWCNT should be given. The surface elemental states of the sample need to be provided, this can be made through XPS survey spectra and high-resolution of C, Fe, O elements.
Answer: It is possible that additional high-resolution photographs would embellish the paper, but unfortunately we do not have access to HRTEM. On the other hand, it seems to us that this information will have no impact on the main problem of studying the role of the magnetic field in topochemical reactions.
3. Remark: As mentioned above, the resolution and layout of Figure 7 should be changed and enhanced. The legends are too general and should be more specific, the diagram sequence is not appropriate.In addition, I wonder how the particle size distribution is measured? It appears to be inconsistent with TEM images?
Answer: We are grateful to the reviewer for this typo. In the new version of the paper the caption of figure 7 has been changed and expanded to explain the meaning of the information. The definition of the dimensional distribution is described in section 2.4 - we used "Image-Pro Plus 6.0" software.
4. Remark: The result discussion section is not specific and too rough.
Answer: We have tried to take the reviewer's comment into account and tried to extend the discussion of the results a bit.
Reviewer 3 Report (New Reviewer)
In my opinion, your article is well described, your results are well discussed and supported by experimental data.
I have only one general comment regarding Figure 5: I observed that the nanocomposite sample is not fully adsorbed on the MWCNT and your strategy is to change the electronic structure of the magnetite nanoparticles, due to their interaction with the MWCNT...could you add some comments on that .... if you can improve the interaction...and how?, or if it is due to the TEM sample preparation?
I recommend you also only to check:
-your conclusion, because two sentences at the end are repeated
-the captions of figures 5 and 7 are reversed.
- Report on figure 8, or its caption, the range of values of the applied magnetic field
- for figure 9, to be consistent between the text and the figure... you mentioned 'degree of transformation over time' and 'degree of conversion'... Moreover to add the unit of the magnetic field value and check legend figure 9...
I enjoyed reading this work.
Author Response
The authors are grateful to the reviewer for reading the article and high estimation. Nevertheless the present version of the article has been revised and slightly expanded. All typos and errors identified by the reviewers have been corrected during the final reading. The English language was checked by a professional translator.
The following changes to the text were made according to the individual comments, which are marked with colour highlighting.
Reviewer #3.
Remark: In my opinion, your article is well described, your results are well discussed and supported by experimental data.
Answer: We are grateful to the reviewer for his high estimation and made recommendations.
1. Remark: I have only one general comment regarding Figure 5: I observed that the nanocomposite sample is not fully adsorbed on the MWCNT and your strategy is to change the electronic structure of the magnetite nanoparticles, due to their interaction with the MWCNT...could you add some comments on that .... if you can improve the interaction...and how?, or if it is due to the TEM sample preparation?
Answer: We appreciate the reviewer's question. We hypothesize that the effect of a magnetic field on reduction reactions can be related to both spin features of the material and charge features. We previously studied the reduction of pure oxides and showed that a magnetic field can accelerate the reaction. If the magnetite particles are part of the composite, the environment can weaken the effect of the magnetic field or, conversely, accelerate it. Unfortunately, at this stage there is no evidence to suggest an effect of carbon nanotubes on reduction reactions. but we plan to do further research in this direction.
2. Remark: I recommend you also only to check your conclusion, because two sentences at the end are repeated
Answer: We are grateful to the reviewer for his careful reading of the text. We have tried to remove duplicate statements in the discussion and conclusions.
3. Remark: the captions of figures 5 and 7 are reversed.
Answer: Thank you, the captions are corrected now.
4. Remark: Report on figure 8, or its caption, the range of values of the applied magnetic field
Answer: Thank you, the figure is corrected now.
5. Remark: - for figure 9, to be consistent between the text and the figure... you mentioned 'degree of transformation over time' and 'degree of conversion'... Moreover to add the unit of the magnetic field value and check legend figure 9...
Answer: Thank you, the figure and caption are corrected now.
Round 2
Reviewer 1 Report (New Reviewer)
no further comments
Author Response
The authors are grateful to the reviewer for reading the article and positive estimation.
Reviewer 2 Report (New Reviewer)
No necessary and suggested Figures and data were added. Although the main purpose of the work is not just to obtain Fe3O4/MWCNT complex catalyst just as the author explained,the structural and morphology information is indeed of great importance to evaluate and correlate the performance. Thus,the manuscript still need to be improved at this stage.
Author Response
The authors are grateful to the reviewer for re-reading the article and for his/her comments. The present version of the article has been slightly revised. Unfortunately, the main experimental data are already included in the text and we can't add anything new. However, we agree with the reviewer that "structural and morphological information is indeed of great importance" and have expanded the text to emphasise this fact.
The following changes have been made to the text (all changes are marked by yellow in the attached file):
The introduction (lines 38-42)
The rate of topochemical reactions, among other things, depends significantly on both the particle size and shape, as well as the defectiveness of the particle surface. The latter determines the number of potential nucleation centres of a new phase. Both the
particle size and its defectiveness are determined by the way of nanoparticle synthesis, their environment, and may vary over a wide range
3.3. Magnetic Characterization (lines 184-188)
It should be noted that the magnetic properties of particles can change considerably when their shape is changed, since the shape anisotropy energy is comparable in magnitude to the magnetocrystalline anisotropy. But this question requires a separate consideration and we hope to show its role in our further investigations
4. Discussion (lines 288-291)
The aim of this work was to study the effect of an external magnetic field on the kinetics of hydrogen reduction of magnetite in a nanocomposite comprising a polymeric matrix in which magnetite nanoparticles immobilised on carbon nanotubes are dispersed.
5. Conclusions (lines 309-311; 319-322)
For the first time we investigated the effect of an external magnetic field on the reduction kinetics not of free magnetite but of magnetite nanoparticles, which are part of the nanocomposite material.
Unfortunately, the question of the effect of nanocomposite morphology, which is undoubtedly important, has been left out of
consideration. However, it requires a separate, serious study and we hope to show this in our future research.
We hope that made corrections will improve the manuscript and it will be recommended for publication.
This manuscript is a resubmission of an earlier submission. The following is a list of the peer review reports and author responses from that submission.
Round 1
Reviewer 1 Report
1) The pictures of the paper should be modified to improve the reader's perception.
2) The paper should add the microscopic characterization diagram of materials, such as SEM diagram.
3) English language and grammar need to be improved。
Author Response
We thank the reviewer for the reading of the manuscript and for the made remarks. We corrected the text according the made remarks.
Reviewer #1 remarks: 1) The pictures of the paper should be modified to improve the reader's perception.
Response. We thank the reviewer for the made remark. We added the pages in the end of the document with separate figures of a better quality.
Reviewer #1 remarks: 2) The paper should add the microscopic characterization diagram of materials, such as SEM diagram.
Response. We are grateful to the reviewer for this recommendation. A section on electron microscopy has been added to the text of the article and a corresponding figure 2 is provided.
Reviewer #1 remarks: 3) English language and grammar need to be improved。
Response. We are grateful to the reviewer for this recommendation. According to the advice the text was checked by our colleague from abroad.
Reviewer 2 Report
The current work focuses on the Effect of an external magnetic field on the hydrogen reduction of magnetite nanoparticles in a polymer matrix. The author’s some effort into the manuscript. Extensive editing of the English language and style is required
Abstract
The main idea wasn’t clear in the abstract, please rephrase it to clear the novelty of this work and the main output
Keywords
Instead of two keywords (nanoparticles and magnetite), they could be merged to be (magnetite nanoparticles)
Introduction
- The introduction is very general and does not provide sufficient background, and all relevant references are not included.
- The novelty of this work is not highlighted and it was not clear the author's contribution in comparison to other previous works.
- Recent review on the magnetic polymer could support the rewriting of the introduction
Magnetic Polymers for Magnetophoretic Separation in Microfluidic Devices. Magnetochemistry. 2021; 7(7):100.
Polymeric Nanocomposites for Environmental and Industrial Applications. International Journal of Molecular Sciences. 2022; 23(3):1023
Experimental
- All the used materials with their impurities should list.
- Experimental methods should be written in detail for easier reproducible by the reader
- Line 57, amount of MWCNT used for the preparation of Fe3O4/MWCNT nanomaterial
- Line 56, what amount of a solution of ammonium hydroxide was used?
- with intense stirring at 15 °C! What is the speed of stirring? How to control this temperature at 15 °C? Did use a closed or open system? An inert gas used?
- Line 60, What do you mean by without pre-drying,!! Is the obtained Fe3O4/MWCNT nanocomposite not washing and not removing impurities or any excess ammonia residue?
- How to separate the product and washing of Fe3O4/MWCNT/PAMMP
- What are the scene rate and range for X-ray diffraction study?
Results
- One of the main problems in the manuscript is that the authors show only results without any interpretations of it or confirmation by citation. More details are required to explain the obtained results.
- Also, it should clear the author's contribution in comparison to other previous works during the discussion.
- First of all Fe3O4/MWCNT/PAMMP nanocomposite should the prepped and confirmed: IR is required to confirm the presence of polymer and iron oxide in the matrix. SEM or TEM is required to show the morphology, size, and particle distribution
- In Figure 1. XRD of Fe3O4/MWCNT/PAMMP, how you investigate the average particle size for magnetite only!!!!
- The peaks are sharp and intense!!! So no effect of the polymer as PAMMP or materials MWCNT as on the magnetite peaks!
- Reference No.17 was used to support your XRD interpretation! But the position of all peaks and indexed peaks are not as in the reference and not in the same range!!!
- VSM figure with Zoom around zero is required to clear the coercive force and the residual magnetization
- No information in the text on the obtained magnetization amount
- Line 124, The saturation magnetization MS of Fe3O4 was accepted as 90 emu/cm3 at ambient conditions, this claim needs a citation
- Line 109-111, More details are required to explain this sentence (The dependence of the relative magnetization on the H/T ratio for two values of T (T=289 K and 390 K) was obtained. Dependence is shown in Fig. 3. The coincidence of the curves confirms the presence of superparamagnetism.)
- Redraw Figures with good resolution and appearance to the reader is required
Conclusion
to form a porous carbon matrix!!!! Where this analysis confirms this claim!!!!! Also no information in the manuscript related to this claim!!
Author Response
The current work focuses on the Effect of an external magnetic field on the hydrogen reduction of magnetite nanoparticles in a polymer matrix. The author’s some effort into the manuscript. Extensive editing of the English language and style is required
We thank the reviewer for the careful reading of the manuscript and for the made remarks. We corrected the text according the made remarks. All comments were considered with the careful explanation. We hope that the corrected manuscript will be acceptable for the publication.
Reviewer #2 remarks: Abstract .
The main idea wasn’t clear in the abstract, please rephrase it to clear the novelty of this work and the main output
Response. We thank for the made remark. The main idea of the research was to investigate the effect of magnetic field on the metallic particle reduction from oxide in case of nanocomposites in situ. So, we rewrote the abstract to make the purpose clearer.
Reviewer #2 remarks: Keywords
Instead of two keywords (nanoparticles and magnetite), they could be merged to be (magnetite nanoparticles)
Response. The Keywords are changed according to the reviewer advice - instead of two keywords we used the expression "magnetite nanoparticles".
Reviewer #2 remarks: Introduction
- The introduction is very general and does not provide sufficient background, and all relevant references are not included.
Response. We are grateful to the reviewer for the circumstances noted. But we would like to draw attention to the fact that a feature of our study is the control of the magnetization of reagents in the recovery process. This method is not applicable to all nanoparticles, but only to those with magnetic ordering at the reaction temperature (these are iron, cobalt and, partially, nickel). There are very few published works on this topic and references to the main works are given [11,12].
- The novelty of this work is not highlighted and it was not clear the author's contribution in comparison to other previous works.
Response. The main difference between our work and others is noted in the response to the previous remark. A feature of this study is the analysis of the features of the recovery of a nanocomposite containing iron oxide and carbon nanotubes. To highlight this fact, the last sentence of the introduction has been rewritten.
- Recent review on the magnetic polymer could support the rewriting of the introduction
Magnetic Polymers for Magnetophoretic Separation in Microfluidic Devices. Magnetochemistry. 2021; 7(7):100.
Polymeric Nanocomposites for Environmental and Industrial Applications. International Journal of Molecular Sciences. 2022; 23(3):1023
Response. We are grateful to the reviewer for the recommendation. These works are included in the list of cited literature with numbers 4 and 5.
Reviewer #2 remarks: Experimental
- All the used materials with their impurities should list.
Response. We are grateful to the reviewer for the recommendation. The section 2.1 was corrected to mark the impurities for all materials.
- Experimental methods should be written in detail for easier reproducible by the reader
Response. We agree with reviewer but need to note that most of experiments were made with the standard equipment which is described. As about the hade-made experimental setup for magnetic measurements we inserted the reference to the paper 11 where the full description is presented.
- Line 57, amount of MWCNT used for the preparation of Fe3O4/MWCNT nanomaterial
Response. Dear reviewer, the carbon nanotubes mass is written as a percent from the weight of ADMPC, but we repeated the value in g-that was equal to 0.000342g.
- Line 56, what amount of a solution of ammonium hydroxide was used?
Response. Dear reviewer, the quantity of the ammonium hydroxide was two times more than mass of two other reagents but we added the quantity - (3.21g).
- with intense stirring at 15 °C! What is the speed of stirring? How to control this temperature at 15 °C? Did use a closed or open system? An inert gas used?
Response. Dear reviewer, we used the automatic stirrer with a thermostat in a fume hood. The speed regulator was near the maximum position. The procedure and setup were descibed in the reference 17.
- Line 60, What do you mean by without pre-drying,!! Is the obtained Fe3O4/MWCNT nanocomposite not washing and not removing impurities or any excess ammonia residue?
Response. Dear reviewer, thank you for this remark! The phrase "without pre-drying" has been removed from the text.
- How to separate the product and washing of Fe3O4/MWCNT/PAMMP
Response. We are grateful to the reviewer for this question. The corresponding information was added to the text: When the synthesis was completed, the mixture was precipitated in a fivefold excess of distilled water. The resulting product was filtered off, washed repeatedly with distilled water to remove residual amounts of reagent, and dried over KOH under vacuum to constant weight. .
- What are the scene rate and range for X-ray diffraction study?
Response. Dear reviewer, thank you for this remark! We add the necessary information to the section 2.3.3 - The diffraction patterns of powder samples were recorded in the Bragg– Brentano focusing geometry with a scan step 0.02 degrees, scan rate 0.01 s/step in the angle range 2theta 20—120degree.
Reviewer #2 remarks: Results
- One of the main problems in the manuscript is that the authors show only results without any interpretations of it or confirmation by citation. More details are required to explain the obtained results.
- Also, it should clear the author's contribution in comparison to other previous works during the discussion.
Response. We agree with the reviewer's comments about the insufficient discussion of the presented results. But at the same time, we believe that in this section it is desirable to describe the main features of the results obtained. And we decided to put the comparison of the described results with the previous ones and the data of other authors in the "Discussion" section. To do this, we added a small introduction in it and also added links to new works 21-23.
- First of all Fe3O4/MWCNT/PAMMP nanocomposite should the prepped and confirmed: IR is required to confirm the presence of polymer and iron oxide in the matrix. SEM or TEM is required to show the morphology, size, and particle distribution
Response. We are grateful to the reviewer for this recommendation. A section on electron microscopy has been added to the text of the article and a corresponding figure 2 is provided. Unfortunately, due to the limited preparation time for the correction of the article, it is not possible to calculate the dimensional distributions of particles. As for the presence of iron oxide, it is confirmed by magnetometry data, which are described in the corresponding section. All other components of the nanocomposite do not detect the field dependence of the magnetic moment.
- In Figure 1. XRD of Fe3O4/MWCNT/PAMMP, how you investigate the average particle size for magnetite only!!!!
Response. We are grateful to the reviewer for the comment made. According to XRD data, the nanocomposite matrix is X-ray amorphous, so only reflexes corresponding to magnetite particles are observed. According to the parameters of these reflexes (position and half-width), the size was estimated, as described in Section 3.1.
- The peaks are sharp and intense!!! So no effect of the polymer as PAMMP or materials MWCNT as on the magnetite peaks!
Response. We are grateful to the reviewer for the comment made. Indeed, the nanocomposite matrix does not affect the crystal structure of magnetite particles. But the presence of carbon nanotubes can affect the electronic structure of magnetic particles, and therefore the kinetics of reduction.
- Reference No.17 was used to support your XRD interpretation! But the position of all peaks and indexed peaks are not as in the reference and not in the same range!!!
Response. We are grateful to the reviewer for the comment made. But it should be noted that in Article 17, for curve (4) with a high iron content, the spectrum fully corresponds to what is shown in Figure 1. Perhaps the opinion about the discrepancy of the data arose due to the poor quality of the figures.
- VSM figure with Zoom around zero is required to clear the coercive force and the residual magnetization
Response. We are grateful to the reviewer for this comment. Unfortunately, the sensitivity of the device does not allow us to insist strictly on the zero value of coercivity and remnant magnetization. In order to avoid ambiguous understanding, this sentence has been corrected: Figure 3 shows the dependence of the magnetization on the field at room temperature, from which it follows that the coercive force and the residual magnetization are near zero.
- No information in the text on the obtained magnetization amount
Response. We are grateful to the reviewer for this remark. The corresponding information was added to the text: The measurement results give a value of 46 emu/g for the saturation magnetization of the sample.
- Line 124, The saturation magnetization MS of Fe3O4 was accepted as 90 emu/cm3 at ambient conditions, this claim needs a citation
Response. We are grateful to the reviewer for this remark. The corresponding reference was added to the text: The saturation magnetization MS of Fe3O4 was accepted as 90 emu/cm3 at ambient conditions [20].
- Line 109-111, More details are required to explain this sentence (The dependence of the relative magnetization on the H/T ratio for two values of T (T=289 K and 390 K) was obtained. Dependence is shown in Fig. 3. The coincidence of the curves confirms the presence of superparamagnetism.)
Response. We are grateful to the reviewer for this comment. The careful consideration of the used approach was made in ref.20. We applied the corresponding relation to our experimental date and wrote below the figure 4 (in previous version - 3). The temperatures were selected as initial state (289K) and reaction temperature (390K).
- Redraw Figures with good resolution and appearance to the reader is required
Response. We thank the reviewer for the made remark. We added the pages with separate figures of a better quality.
Reviewer #2 remarks: Conclusion
to form a porous carbon matrix!!!! Where this analysis confirms this claim!!!!! Also no information in the manuscript related to this claim!!
Response. We are grateful to the reviewer for this remark and sorry for the made mistake. The wrong sentence was deleted from the text.

Round 2
Reviewer 2 Report
Accept in the present form